# Revealing and Protecting Labels in Distributed Training

**Trung Dang**[*]
Boston University
trungvd@bu.edu

**Om Thakkar**
Google
omthkkr@google.com

**Swaroop Ramaswamy**
Google
swaroopram@google.com

**Rajiv Mathews**
Google
mathews@google.com

**Peter Chin**
Boston University
spchin@bu.edu

**Françoise Beaufays**
Google
fsb@google.com

## Abstract

Distributed learning paradigms such as federated learning often involve transmission of model updates, or gradients, over a network, thereby avoiding transmission of private data. However, it is possible for sensitive information about the training data to be revealed from such gradients. Prior works have demonstrated that labels can be revealed analytically from the last layer of certain models (e.g., ResNet), or they can be reconstructed jointly with model inputs by using Gradients Matching [1] with additional knowledge about the current state of the model. In this work, we propose a method to discover the set of labels of training samples from only the gradient of the last layer and the id to label mapping. Our method is applicable to a wide variety of model architectures across multiple domains. We demonstrate the effectiveness of our method for model training in two domains - image classification, and automatic speech recognition. Furthermore, we show that existing reconstruction techniques improve their efficacy when used in conjunction with our method. Conversely, we demonstrate that gradient quantization and sparsification can significantly reduce the success of the attack.

## 1 Introduction

Distributed learning paradigms such as federated learning [2] offer a mechanism for mobile devices to collaboratively train deep learning models via transmission of model updates over the network, while ensuring sensitive training data remains resident, locally, on device. Recent works have demonstrated that information about training data can be revealed from a gradient, or a model update in general. For instance, in the image classification domain, the leaked information ranges from some property of the data like *membership inference* [3, 4], to *pixel-wise* accurate image reconstruction [1]. To reconstruct the samples given to a model, Zhu et al. [1] propose Deep Leakage from Gradients (DLG), or Gradients Matching (GM), a method that iteratively updates a dummy training sample to minimize the distance of its gradient to a target gradient sent from a client. This has been used to successfully reconstruct large images in a mini-batch [5, 6, 7]. The application of GM has also been demonstrated in other domains, such as language modeling [1] and automatic speech recognition [8].

As opposed to model inputs being continuous, training labels being *discrete* prevents common techniques like gradient descent from being trivially employed for reconstructing them. Instead, training labels can be reconstructed jointly with model inputs by optimizing for continuous "pseudo" labels (a probability distribution on labels) [1]. However, this method is not guaranteed to succeed,

---

[*]Work performed while at Google.

35th Conference on Neural Information Processing Systems (NeurIPS 2021).

for instance, when the weight update is computed from a single sample. Zhao et al. [9] first observe that a label of a single training image can be analytically derived from a gradient with 100% success rate. Their method, however, only applies to gradients computed from a single sample, and does not apply to gradients computed from a mini-batch, or aggregated from multiple training steps. Yin et al. [7] and Wainakh et al. [10] extend this method to demonstrate that labels of samples in a mini-batch can be recovered. The method leads to high success rate on revealing labels of samples in the batch, but its applicability is restricted to classification models using a non-negative activation function before the fully-connected layer.

While many works have focused on sample reconstruction, the knowledge of training labels has been shown to be necessary for high-quality reconstruction [7]. In sequence models (e.g., speech and language domains), revealing the labels itself can result in a significant leak of privacy. Speech transcript leakage imposes a significant threat to users' privacy and can lead to the exposure of speaker identities [8]. Text-form data may include secrets such as passwords, social security numbers, or sensitive words that reveal the context of the conversation. However, in sequence models that output natural language, the number of labels (e.g., characters, wordpieces, or words) can be substantially larger than in image classification models. For such large search spaces, the joint optimization approach adopted in [1] becomes ineffective.

In this work, we make the following contributions.

1. We propose a method to reveal the entire set of labels used in computing a weight update. We refer to the method as Revealing Labels from Gradients (RLG). The method is applicable with any model that uses a softmax activation with cross-entropy loss, thus is applicable to a wide variety of deep learning architectures. We empirically demonstrate that RLG can be applied to a parameter update generated from a mini-batch (1-step, $N$-sample), or after several steps ($K$-step, 1-sample each), or even the most general case of $K$-step update of $N$-sample at each step.

2. We discuss the effects of gradient quantization and sparsification techniques, namely Sign-SGD and GradDrop, in the prevention of RLG. These methods have been used to reduce the communication cost in distributed training. We discover that they are also effective in mitigating RLG.

3. We further demonstrate that in sequence models, the knowledge about the set of labels can be used to fully reconstruct a sequence of labels in order, while GM has limited success. In addition to the parameter update, this requires the current state of the model.

We demonstrate our method on 1) an image classification task and 2) an automatic speech recognition (ASR) task. For image classification, we present our results on ResNet [11], which has often been the target of choice for attacks in previous works [5, 7], and EfficientNet [12], the state-of-the-art architecture for image classification. While the approach proposed in [7] works for the ResNet, only our proposed method works for EfficientNet. For ASR, we demonstrate our method with the attention-based encoder-decoder architecture. We show that our method used in conjunction with GM can perform full transcript reconstruction, while none of prior works succeed. Our code is published at `https://github.com/googleinterns/learning-bag-of-words`.

## 2 Revealing Training Labels from Gradients

### 2.1 Background & Notation

We consider a scenario of model training where an adversary gets access to some model updates (e.g., gradients). This can be possible in distributed learning settings by honest-but-curious participants, or a honest-but-curious server orchestrating the training, or an adversary that compromises some client-to-server communication channel. Note that we do not cover distributed training paradigms that share an intermediate representation, since they usually involve disclosing labels to the training agent [13] and require different types of defense techniques [14]. For simplicity, we consider vanilla gradient sharing approach, and assume that no label obfuscation is performed on the client side.

We consider model architectures having a softmax activation with cross-entropy loss for classification, which is standard in deep learning based architectures (e.g. [11, 15]). The goal is to reveal labels $y$

for labeled samples of the form $(\boldsymbol{X}, \boldsymbol{y})$. For each training sample, $\boldsymbol{y}$ could be a single label (e.g., in classification models), or a sequence of labels (e.g., in sequence models).

Let $\boldsymbol{W} \in \mathbb{R}^{d \times C}$ and $\boldsymbol{b} \in \mathbb{R}^C$ be the weight and bias of the layer prior to the softmax function (referred to as the projection layer), where $d$ is the input dimension (same as output dimension of the previous layer), and $C$ is the number of classes. The projection layer maps the latent representation $\boldsymbol{h} \in \mathbb{R}^{1 \times d}$ extracted from the model input to $\boldsymbol{z} = \boldsymbol{h}\boldsymbol{W} + \boldsymbol{b} \in \mathbb{R}^{1 \times C}$, which represents the unnormalized log probabilities (logits) on the classes.

We assume that the adversary can access $\Delta \boldsymbol{W}$, i.e. the weight update of the projection layer, computed using back-propagation. This could be computed from a single sample, a batch of several samples, or several batches. Note that our setting crucially does not require the adversary to have access to any other part of the update except $\Delta \boldsymbol{W}$; however, accessing to the full model update can be useful for sequence reconstruction, as discussed in Section 4. Additionally, the adversary is required to know the id to label mapping, so that columns in $\Delta \boldsymbol{W}$ can be associated with actual labels. For the target of the method, we aim to reveal a set of labels from samples used to compute $\Delta \boldsymbol{W}$, i.e., $\mathcal{S} = \{y \mid y \in \boldsymbol{Y}\}$. The method does not return the number of samples for each class.

In this section, we show that $\Delta \boldsymbol{W}$ can be represented as the product of two matrices (Remark 1), one of which can be related to $\boldsymbol{Y}$ (Remark 3). Based on this property, we propose a method using singular value decomposition (SVD) to reveal the set of labels $\mathcal{S}$ given access to $\Delta \boldsymbol{W}$.

## 2.2 Gradient of the Projection Layer

Before explaining our method, we want to derive some properties of $\Delta \boldsymbol{W}$. We first consider the case when the gradient is computed from a single sample with a single label $(\boldsymbol{X}, y)$. The probability distribution $\hat{\boldsymbol{y}}$ over the classes is derived by applying the softmax function on $\boldsymbol{z}$: $\hat{y}_i = \frac{\exp(z_i)}{\sum_j \exp(z_j)}$. Training a classification model involves minimizing the cross-entropy loss

$$\mathcal{L} = -\sum_i [y = i] \log \hat{y}_i = -\log \frac{\exp z_{y_c}}{\sum_{j \in \mathcal{C}} \exp z_j} \tag{1}$$

Since $\boldsymbol{z} = \boldsymbol{h}\boldsymbol{W} + \boldsymbol{b}$, we have $\frac{\partial \boldsymbol{z}}{\partial \boldsymbol{W}} = \boldsymbol{h}^\top$. The weight update for the projection layer can be represented as

$$\Delta \boldsymbol{W} = \frac{\partial \mathcal{L}}{\partial \boldsymbol{W}} = \frac{\partial \mathcal{L}}{\partial \boldsymbol{z}} \frac{\partial \boldsymbol{z}}{\partial \boldsymbol{W}} = \boldsymbol{h}^\top \boldsymbol{g}, \qquad \text{where } \boldsymbol{g} = \frac{\partial \mathcal{L}}{\partial \boldsymbol{z}} \tag{2}$$

(Note that $\boldsymbol{h}$ and $\boldsymbol{g}$ are row vectors). We can generalize this property to the following settings, which involve more than one training samples and labels.

**Weight update by a mini-batch / sequence of labels** The weight update of a $N$-sample mini-batch or a sequence of length $N$ is averaged from weight updates computed from each sample in the batch or each label in the sequence: $\Delta \boldsymbol{W} = \frac{1}{N} \sum_{i=1}^N \boldsymbol{h}_i^\top \boldsymbol{g}_i = \boldsymbol{H}^\top \boldsymbol{G}$, where $\boldsymbol{H} = \frac{1}{N}[\boldsymbol{h}_1, \ldots, \boldsymbol{h}_N]$, and $\boldsymbol{G} = [\boldsymbol{g}_1, \ldots, \boldsymbol{g}_N]$

**Weight update by several training steps** The weight update after $K$ training steps is the sum of weight update at each step[2]: $\Delta \boldsymbol{W} = \sum_{i=1}^K \alpha_i \Delta \boldsymbol{W}_{(i)} = \sum_{i=1}^K \alpha_i \boldsymbol{H}_{(i)}^\top \boldsymbol{G}_{(i)} = \boldsymbol{H}^\top \boldsymbol{G}$, where $\Delta \boldsymbol{W}_{(i)}$ and $\alpha_i$ is the softmax gradient and learning rate at the time step $i$, respectively, $\boldsymbol{H} = [\alpha_1 \boldsymbol{H}_{(1)}, \ldots, \alpha_K \boldsymbol{H}_{(K)}]$, and $\boldsymbol{G} = [\boldsymbol{G}_{(1)}, \ldots, \boldsymbol{G}_{(K)}]$. More generally, we have the following.

**Remark 1.** $\Delta \boldsymbol{W}$ *can be represented as the product of* $\boldsymbol{H}^\top \in \mathbb{R}^{d \times S}$ *and* $\boldsymbol{G} \in \mathbb{R}^{S \times C}$*, where* $S$ *is the number of labels used to compute the weight update.*

We consider $S$ unknown to the adversary. If the weight update is computed from a batch, $S$ is the batch size. If the weight update is aggregated from several training steps, $S$ is the total number of samples at these steps. In fact, $S$ can be inferred from the rank of $\Delta \boldsymbol{W}$ as follows.

---

[2]We consider non-adaptive optimization methods, e.g. FedAvg [16], which are common in FL training.

**Remark 2.** *Assume that $S < \min\{d, C\}$ and $\boldsymbol{H}$, $\boldsymbol{G}$ are full-rank matrices. We have $\text{rank}(\boldsymbol{H}) = \text{rank}(\boldsymbol{G}) = S$, and $S$ can be inferred from $S = \text{rank}(\Delta \boldsymbol{W})$.*

Now we want to derive a relation between $\boldsymbol{G}$ and labels $\boldsymbol{Y}$. Differentiating $\mathcal{L}$ w.r.t. $\boldsymbol{z}$ yields

$$g_i^j = \nabla z_i^j = \frac{\partial \mathcal{L}}{\partial z_i^j} = \begin{cases} -1 + \text{softmax}(z_i^j, \boldsymbol{z}_i) & \text{if } j = y_i \\ \text{softmax}(z_i^j, \boldsymbol{z}_i) & \text{otherwise} \end{cases} \tag{3}$$

Since the softmax function always returns a value in $(0, 1)$, each row in $\boldsymbol{G}$ has a unique negative coordinate corresponding to the ground-truth label. Formally, we have

**Remark 3.** *Let $Neg(\boldsymbol{u})$ define the indices of negative coordinates in a vector $\boldsymbol{u}$. For every row $\boldsymbol{g}_i$ in $\boldsymbol{G}$, $Neg(\boldsymbol{g}_i) = \{y_i\}$.*

More intuitively, in order to minimize the loss, the probability of the ground-truth label should be pushed up to 1 (by a negative gradient), and probabilities of other labels should be pushed down to 0 (by a positive gradient). Hence, signs in the gradient matrix $\boldsymbol{G}$ can be directly linked to labels $\boldsymbol{Y}$. Though $\boldsymbol{G}$ is not directly observable from $\Delta \boldsymbol{W}$, this property of $\boldsymbol{G}$ is critical for revealing the set of labels from $\Delta \boldsymbol{W}$, as explained in the next section.

### 2.3  Proposed Method

In this section, we propose a method to reveal the set of labels $\mathcal{S}$ from $\Delta \boldsymbol{W}$, which we refer to as Revealing Labels from Gradients (RLG). For our method to work, we need to assume that $S < \min\{d, C\}$, i.e., the method works when $\Delta \boldsymbol{W}$ is aggregated from the weight updates computed from a reasonable number of samples. This is justifiable, since when running distributed training on private data on edge devices, it is a good practice to use a small batch size [17]. For sequence models that output natural language, $d$ and $C$ are usually in the order of thousands or tens of thousands, which is often much larger than $S$.

By the singular value decomposition, $\Delta \boldsymbol{W}$ can be decomposed into $\boldsymbol{P}\boldsymbol{\Sigma}\boldsymbol{Q}$, where $\boldsymbol{P} \in \mathbb{R}^{d \times S}$ and $\boldsymbol{Q} \in \mathbb{R}^{S \times C}$ are orthogonal matrices, and $\boldsymbol{\Sigma} \in \mathbb{R}^{S \times S}$ is a diagonal matrix with non-negative elements on the diagonal. The following remark is a corollary of (3).

**Remark 4** (Existence of a hyperplane). *Assume that there is a sample with label c. There exists a vector $\boldsymbol{r} \in \mathbb{R}^S$ such that $\boldsymbol{r}\boldsymbol{q}^c < 0$ and $\boldsymbol{r}\boldsymbol{q}^{j \neq c} > 0$, or $Neg(\boldsymbol{r}\boldsymbol{q}) = \{c\}$, where $\boldsymbol{q}^j$ is the j-th column in $\boldsymbol{Q}$. In other words, the point $\boldsymbol{q}^c$ can be separated from other points $\boldsymbol{q}^{j \neq c}$ in the S-dimensional space by a hyperplane $\boldsymbol{r}\boldsymbol{x} = 0$.*

If a label $c$ appears in the batch, we have $y_i = c$ for some $i$, or $Neg(\boldsymbol{g}_i) = \{c\}$. Let $\boldsymbol{r} = \boldsymbol{g}_i \boldsymbol{Q}^\top$, then we have $\boldsymbol{r}\boldsymbol{Q} = \boldsymbol{g}_i$, or $Neg(\boldsymbol{r}\boldsymbol{Q}) = \{c\}$. Therefore, if a label $c$ appears in the batch, there exists a linear classifier without bias that separates $\boldsymbol{q}^c$ from $\boldsymbol{q}^{j \neq c}$. The problem of finding a perfect classifier can be solved via linear programming: if there exists a classifier that separates $q^c$ from $q^{j \neq c}$, the following problem has a solution.

$$\text{LP}(c): \min_{\boldsymbol{r} \in \mathbb{R}^N} \quad \boldsymbol{r}\boldsymbol{q}^c \qquad \text{s.t.} \qquad \boldsymbol{r}\boldsymbol{q}^c \leq 0 \qquad \text{and} \qquad \boldsymbol{r}\boldsymbol{q}^j \geq 0, \forall j \neq c \tag{4}$$

Note that this defines a necessary, but not sufficient condition for $c$ being used to compute the model update. Figure 1 and Algorithm 1 provide an illustration and pseudo-code for our approach. In Figure 1, the points $q_2$ and $q_5$ are separable from the rest by a hyperplane passing through the origin, hence they are predicted as a label in $\mathcal{S}$. The other points, however, are not separable. From the remark 4, it is certain that they are not included in $\mathcal{S}$.

### 2.4  Comparison with Related Works

There have been efforts to look at the gradient of the projection layer to infer information about training labels. Zhao et al. [9] are the first to observe that a training label can be retrieved from $\Delta \boldsymbol{W}$. However, they provide a method that works only with a gradient computed from a single data sample. The method is based on the observation that $\Delta \boldsymbol{W}^j = \boldsymbol{h}^\top g^j$, thus $\Delta \boldsymbol{W}^i \cdot \nabla \boldsymbol{W}^j = \|\boldsymbol{h}\|^2 g^i g^j$. Since

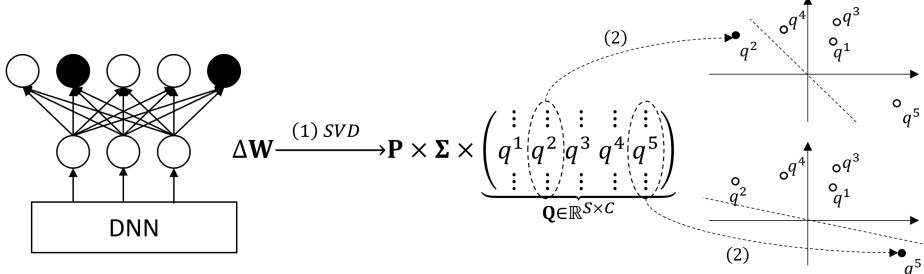

Figure 1: An illustration of RLG. Black nodes correspond to labels of samples used to compute the weight update. (1) The weight update of the projection layer $\Delta W$ is decomposed into $P\Sigma Q$ by SGD. (2) Each column in $Q$ is mapped to a point in an $S$-dimensional space. If a point is separable from the rest, the corresponding label is predicted as being included in $\mathcal{S}$.

---

**Algorithm 1** Reveal the set of labels from a weight update of the projection layer

---

**Input:** Gradient of the projection layer $\Delta W \in \mathbb{R}^{d \times C}$, all labels $\mathcal{C} = \{c_i\}_{i=1}^{C}$
$S \leftarrow \text{rank}(\Delta W)$
Find $Q \in \mathbb{R}^{S \times C}$ the right singular matrix of $\Delta W$
$\mathcal{S} = \varnothing$
**for** $i = 1$ **to** $C$ **do**
  **if** $\text{LP}(c_i)$ has a solution **then**
    Add label $c_i$ into $\mathcal{S}$
  **end if**
**end for**
**Return:** Number of labels $|Y| = S$, set of labels $\mathcal{S}$

---

$g^c < 0$ and $g^{i \neq c} > 0$, the training label $c$ can be obtained by finding a column in $\Delta W$ whose dot product with other columns results in a negative number. Yin et al. [7], on the other hand, assume that $H \geq 0$, i.e. a non-negative non-linear activation function is applied before the projection layer, and demonstrate that labels in a batch can be revealed with high success rate with this assumption. The gradient computed from the batch $\Delta W$ is the super-position of gradients computed from each sample $\Delta W_{(i)}$. Since $\Delta W_{(i)} = h_i^\top g_i$, $\text{Neg}(g_i) = \{y_i\}$, and $h > 0$, the $y_i$-th column in $\Delta W_{(i)}$ has all negative values and other columns have all positive values. They empirically observe that the magnitudes of negative columns dominate those of positive columns, thus a negative element $\Delta w_i^j$ in $\Delta W$ hints that there exists a $\Delta W_{(i)}$ with the $j$-th negative column, or the label $j$ appears in the batch. The set of labels in the batch is retrieved by

$$\mathcal{S} = \left\{ j \mid \min_{i \in [d]} \Delta w_i^j < 0 \right\} \tag{5}$$

Without the $H \geq 0$ assumption, Yin et al.'s method does not work. For $H$ to be a non-negative matrix, this approach requires that non-negative activation functions being applied prior to the input of the projection layer. While non-negative activation functions such as ReLU are widely used in an image classification model, other variations such as Leaky RELU [18], or elu [19] may return negative values as to address the dying ReLU problem. In sequence modeling, the hyperbolic tangent function ($\tanh$) is usually employed, for example, in LSTM [20], or attention mechanism [21]. Our approach does not assume anything about $H$, which makes it applicable to a variety of deep learning architectures.

However, there are some limitations of RLG. First, we require $S < \min\{d, C\}$ for SVD; therefore, the method is not applicable when the number of samples $S$ is large and the number of classes $C$ is small. Second, since RLG relies on matrix decomposition and linear programming solutions, it is slower than (5), and precision issue could arise when $\|\Delta W\| \approx 0$ (e.g. at a late training stage).

Table 1: Precision, Recall, F1, and EM score of set of labels prediction on ResNet50 and Efficient-NetB0. While Equation (5) from Yin et al. [7] performs comparably with RLG on ResNet50, only RLG succeeds on the EfficientNetB0 model.

| | $N$ | ResNet50 | | | | EfficientNetB0 | | | |
| --- | --- | --- | --- | --- | --- | --- | --- | --- | --- |
| | | Prec | Recall | F1 | EM | Prec | Recall | F1 | EM |
| Yin et al. [7] (Eq (5)) | 10 | 1 | .996 | .998 | .96 | .010 | 1 | .020 | 0 |
| | 50 | 1 | .974 | .987 | .27 | .049 | 1 | .093 | 0 |
| | 100 | 1 | .960 | .979 | .04 | .095 | 1 | .174 | 0 |
| RLG (ours) | 10 | .999 | .998 | .998 | .97 | 1 | 1 | 1 | 1 |
| | 50 | .998 | .976 | .987 | .27 | .999 | 1 | .999 | .95 |
| | 100 | .978 | .966 | .972 | .02 | .998 | .999 | .999 | .78 |

## 2.5 Experiments

We demonstrate our approach on 1) an image classification task, and 2) an automatic speech recognition task. We consider our task as a binary classification task, which returns 1 (positive) if a label is included in the training data and 0 (negative) otherwise. The Precision, Recall, and F1 score can be computed from this setup. We also provide the Exact Match (EM) score to measure the accuracy of the entire set. In the ASR task, we also report the length error (LE, the average distance to the ground-truth length) when inferring the length of transcript $S$ from the remark 2.

### 2.5.1 Image Classification

We demonstrate RLG on ResNet [11], which has been chosen as the target model in previous works [1, 5, 7], and EfficientNet [12], the current state-of-the-art architecture on the ImageNet benchmark [22]. We implement the baseline approach in [7], using the equation (5). For EfficientNet, the model uses the swish activation function [23, 24], which may return a negative value. As discussed in Section 2.4, previous works [9, 7] cannot be adapted to this setting. We randomly sample 100 batches of size $N = 10, 50, 100$ from the validation set of ImageNet. We use the ImageNet pre-trained models provided by Tensorflow (Keras) [25] library. The average sizes of the set of labels for batch of 10, 50, and 100 samples are 10.0, 48.8, and 92.5 labels, respectively.

Table 1 shows the Precision, Recall, F1, and EM score for the baseline and our method. For ResNet50, the baseline method performs well with perfect Precision and near-perfect F1 score. For EfficientNetB0, however, it fails to give the set of labels, while RLG can predict correctly for 95% of the 50-sample batches, and 78% of the 100-sample batches.

### 2.5.2 Automatic Speech Recognition

We apply RLG on the weight update from an encoder-decoder attention-based ASR model [26, 15] to demonstrate its application in revealing the Bag of Words (BoW) of spoken utterances. The model is a word-piece-based model, implemented in lingvo [27] using Tensorflow [25]. The number of word pieces in the vocabulary is 16k. We demonstrate BoW prediction for utterances from LibriSpeech [28]. For each length value of less than or equal to 50 characters, we sample at most 10 utterances from the combined test set of LibriSpeech. The total number of sampled utterances is 402.

Equation (5) [7] cannot be used for this architecture, since a hyperbolic tangent activation function is used in the ASR model. We implement a baseline with Gradients Matching [1] to optimize the one-hot matrix of the transcript jointly with the input to the decoder (context vector), by matching the gradient of the decoder. We run the reconstruction with learning rate 0.05, decaying by a factor of 2 after every 4,000 steps. The reconstruction stops after 2,000 steps that the transcript remains unchanged. The method should be able to reconstruct the full transcript; however, we only report the BoW prediction results. In the baseline, weight update is obtained from an untrained model.

For RLG, we implement BoW prediction on a weight update from an untrained model, a partially trained model (4k steps), and a fully trained model (10k steps). The training is performed on centralized training data, with batch size 96 across 8 GPUs. Since iterating over 16k labels takes time, we perform a screening round. Consider each column in $Q$ as a data point in a $S$-dimensional space.

Using the Perceptron algorithm [29], the screening round simply uses 500 points with the largest norm to filter out all points that are not separable from them. This is faster than solving the full LP problem. Only points that pass the screening round are processed in our algorithm. Each inference takes less than 5 minutes on a CPU.

Table 2 shows LE, Precision, Recall, F1 score of the prediction, along with EM score of the entire BoW. For an untrained model, RLG can recover the exact BoW for 98% utterances. While these results slightly drop on partially trained and trained models, they remain highly accurate with a recall of 1. For the baseline, Gradients Matching fails on most utterances, with only 1.8% EM.

We also demonstrate RLG for a weight update computed from $K$ steps, with a $N$-sample mini-batch at each step, after one or $K$ training steps. Table 3 shows Precision, Recall, F1, and EM score for the cases of multi-sample single-step ($N = 4, 8$), single-sample multi-step ($K = 4, 8$), and multi-sample multi-step ($N = 2, K = 2$). While $S$ can be inferred from $\Delta W$ (remark 2), in practice, when $S$ is large, its prediction is erroneous due to precision issue, especially when multi steps are involved (for example, the average length estimate error for the 8-step case is 8.18). Here we use the ground-truth length for the value of $S$. While RLG achieves perfect score on the multi-sample single-step case, there is a drop when the weight update is computed from several steps.

| | LE | Prec | Rec | F1 | EM |
|---|---|---|---|---|---|
| GM [1] | - | .007 | .008 | .008 | .018 |
| RLG | 0 | .998 | 1 | .999 | .988 |
| RLG(4k) | 0 | .993 | 1 | .996 | .955 |
| RLG(10k) | .092 | .989 | 1 | .994 | .933 |

Table 2: Length Error, Precision, Recall, F1 score of the prediction, along with EM score of the BoW. For RLG, we also run it on a weight update for a partially trained model (4k steps), and a trained model (10k steps).

| $N$ | $K$ | Prec | Rec | F1 | EM |
|---|---|---|---|---|---|
| 4 | 1 | 1 | 1 | 1 | 1 |
| 8 | 1 | 1 | 1 | 1 | 1 |
| 1 | 4 | .959 | .903 | .930 | .535 |
| 1 | 8 | .931 | .862 | .895 | .160 |
| 2 | 2 | 1 | .914 | .955 | .330 |

Table 3: Precision, Recall, F1, and EM score of the prediction when the weight update is computed from a mini-batch of $N$ samples, after $K$ steps.

## 3  Defense Strategies

To defend against our method, distributed training should avoid exchanging precise gradients, so that the linear dependency in (2) does not hold. In this section, we investigate the effect of two common gradient compression techniques, namely gradient quantization (e.g. Sign-SGD) and gradient sparsification (e.g. GradDrop), on RLG.

**Gradient Quantization (e.g. Sign-SGD)**   Using only the sign of gradient for distributed stochastic gradient descent has good practical performance [30, 31], and has been studied in the context of distributed training to reduce the communication cost with nearly no loss in accuracy [32, 33, 34, 35]. This has also been studied in the context of federated learning with theoretical guarantees [36].

**Gradient Sparsification (e.g. GradDrop)**   Distributed training can be more communication-efficient with gradient sparsification techniques. By proposing GradDrop, which simply maps up to 99% smallest updates by absolute value to 0, Aji et al. [37] demonstrate achieving the same utility on several tasks. The compression rate can also be self-adapted for better efficiency [38].

We use Sign-SGD, and GradDrop with the rate of 50% and 90% on the weight update of the projection layer to demonstrate the defensive effects of these techniques on RLG. We report the results of the EfficentNetB0 model and the ASR model for the settings that RLG performs best, as reported in Section 2.5.1 (i.e., the batch size of 10) and Section 2.5.2 (i.e., the untrained model).

**Results**   Table 4 compares the Precision, Recall, F1, and Exact Match score of RLG for model with and without defense techniques. We can see that Sign-SGD can mitigate RLG in most cases, with 0% EM on EfficientNetB0 and 10.4% EM on ASR. GradDrop, however, does not have a significant effect on the success of RLG in some settings, e.g., the ASR model. Since $\Delta W$ may already be sparse, applying GradDrop may have a little effect on breaking the linear dependency (2).

Table 4: Precision, Recall, F1, and EM score of different defense strategies on RLG for weight updates computed from an EfficientNetB0 model, and an ASR model.

| | EfficientNetB0 | | | | ASR | | | |
|---|---|---|---|---|---|---|---|---|
| | P | R | F1 | EM | P | R | F1 | EM |
| No Defense | 1 | 1 | 1 | 1 | .998 | 1 | .999 | .988 |
| Sign-SGD | .010 | 1 | .020 | 0 | .504 | .181 | .266 | .104 |
| GradDrop 50% | .999 | .843 | .914 | .17 | .857 | 1 | .923 | .881 |
| GradDrop 90% | 1 | .294 | .455 | 0 | .997 | .981 | .989 | .866 |

## 4 Reveal Sequences of Labels

In sequence models, the algorithm in Section 2 only returns a BoW. While the knowledge of BoW, in many cases, can be used to reveal the content of an utterance, many sensitive phrases, such as passwords or numbers, may be composed of uncommon word combination. In this section, we propose an approach that utilizes the BoW to reconstruct the full transcript.

### 4.1 Gradients Matching with BoW

We use $\boldsymbol{W}_{\text{model}}$ to denote the weights of the model. Let $(\boldsymbol{X}^*, \boldsymbol{y}^*)$ be the training sample used to compute the gradient of $\boldsymbol{W}_{\text{model}}$. The weight update computed from $(\boldsymbol{X}^*, \boldsymbol{y}^*)$ is $\Delta \boldsymbol{W}^*_{\text{model}} = \frac{\partial \mathcal{L}}{\partial \boldsymbol{W}_{\text{model}}}(\boldsymbol{X}^*, \boldsymbol{y}^*)$. In a sequence model, $\boldsymbol{X}^*$ and $\boldsymbol{y}^*$ have variable lengths. A malicious adversary who wants to reconstruct $\boldsymbol{X}^*$ and $\boldsymbol{y}^*$ via Gradients Matching tries to solve the following optimization:

$$\min_{\boldsymbol{X}, \boldsymbol{y}} \quad \left\| \frac{\partial \mathcal{L}}{\partial \boldsymbol{W}_{\text{model}}}(\boldsymbol{X}, \boldsymbol{y}) - \nabla \boldsymbol{W}^*_{\text{model}} \right\| \qquad \text{s.t.} \qquad \boldsymbol{X} \in \mathbb{R}^{T \times d}, \boldsymbol{y} \in \mathcal{C}^{|S|} \tag{6}$$

where $T$ is the length of model input, $\mathcal{C}$ is the set of labels, and $S$ be the length of the sequence. Besides the model architecture, the current model weights $\boldsymbol{W}_{\text{model}}$, and the weight update $\boldsymbol{W}_{\text{model}}$ computed from a training sample, the adversary is also required to know the length of the model input $T$, and the length of the model output $S$.

Since $\boldsymbol{y}$ is a discrete variable, it cannot be optimized with gradient-based methods. Instead of optimizing for $\boldsymbol{y}$, we can optimize for the post-softmax probability distribution $\boldsymbol{P} \in \mathbb{R}^{S \times C}$, which we refer to as smooth labels. The loss on smooth labels is modified from (1)

$$\mathcal{L} = -\sum_{i=1}^{S} \sum_{k=1}^{C} p_i^k \log \frac{\exp z_i^k}{\sum_{j=1}^{C} \exp z_i^j} \tag{7}$$

Optionally, a regularization term can be added to the objective in (6) to constrain the smooth labels $\boldsymbol{P}$. We use the regularization $\mathcal{R}(\boldsymbol{P}) = \sum_{i=1}^{S} \|\|\boldsymbol{p}_i\| - 1\|$ to enforce that each row in $\boldsymbol{P}$ sums to 1.

The embedding of a regular label $c_i$, i.e., $e_i = E(c_i)$ where $E$ is the lookup dictionary, if required in the network, can be changed to a smooth label embedding $e_i = \sum_{k=1}^{C} p_i^k E(c_k)$. Now let $\mathcal{S} = \{c_{i_1}, c_{i_2}, ..., c_{i_{|\mathcal{S}|}}\}$ denote the BoW obtained from the Algorithm 1 in Section 2. Since if a word is included in the ground-truth transcript, it is guaranteed to be included in $\mathcal{S}$, our search space can be restricted to words included in $\mathcal{S}$. More specifically, let $\boldsymbol{P}' \in \mathbb{R}^{S \times |\mathcal{S}|}$ denote the smooth labels for words included in the BoW. By setting elements in $\boldsymbol{P}$ that are not associated with a word in the BoW to 0, the loss (7) becomes $\mathcal{L}_{BoW} = -\sum_{i=1}^{S} \sum_{k=1}^{|\mathcal{S}|} p_i'^k \log \frac{\exp z_i^{c_{i_k}}}{\sum_{j=1}^{C} \exp z_i^j}$.

With the BoW restriction, the search space for labels is reduced from $S \times C$ to $S \times |\mathcal{S}|$. This is a significant reduction since the size of the vocabulary $C$ is much larger than the size of the BoW for a sentence or an utterance. Note that the optimization task (6) also requires knowledge about the length of the model input $T$, which should not be known by the adversary. In the encoder-decoder

Table 5: Number of variables to optimize, WER, and EM score of transcript reconstruction by GM with and w/o BoW. For GM with BoW, we also repeat the reconstruction up to 5 times and report the best results in terms of gradients distance.

| | #vars | WER | EM |
|---|---|---|---|
| GM w/o BoW (1-run) | 130k | >1 | .018 |
| GM with BoW (1-run) | 20k | .284 | .519 |
| GM with BoW (5-run) | 20k | .010 | .975 |

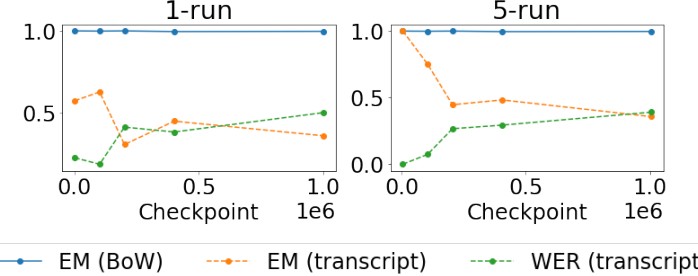

Figure 2: The results for BoW prediction (F1, EM scores) and full transcript reconstruction (EM, WER scores) from a weight update computed by model checkpoints at 0, 1k, 2k, 4k, and 10k steps.

architecture with attention mechanism, we eliminate the requirement for $T$ by optimizing for the context vectors $\boldsymbol{A} \in \mathbb{R}^{S \times d_{\boldsymbol{A}}}$ that is given to the decoder via an attention mechanism, instead of optimizing for $\boldsymbol{X}$. Here, $S$ can be inferred from $\nabla \boldsymbol{W}$ in Algorithm 1, and $d_{\boldsymbol{A}}$ can be inferred from the model architecture. The optimization problem (6) becomes

$$\min_{\boldsymbol{A}, \boldsymbol{P}'} \left( \left\| \frac{\partial \mathcal{L}_{BoW}}{\partial \boldsymbol{W}_{\text{decoder}}} (\boldsymbol{A}, \boldsymbol{P}') - \nabla \boldsymbol{W}^*_{\text{decoder}} \right\| + \lambda \mathcal{R}(\boldsymbol{P}') \right) \quad \text{s.t.} \quad \boldsymbol{A} \in \mathbb{R}^{S \times d_{\boldsymbol{A}}}, \boldsymbol{P}' \in \mathbb{R}^{S \times |\mathcal{S}|} \quad (8)$$

where $\lambda$ is the weight of the regularization term.

## 4.2 Experiments

We conduct full transcript reconstruction on 402 utterances sampled in Section 2.5.2, with and without BoW, by minimizing the objective function in (8) with $\lambda = 1$. We start with learning rate 0.05, reducing by half after every 4,000 steps until it reaches 0.005. The reconstruction stops when the transcript remains unchanged after 2,000 steps. The reconstruction operates on a gradient computed from a few-step trained model. We also run the reconstruction up to five times for GM with BoW and report the results of runs with the lowest gradients distance. The average running time for each reconstruction ranges from 10 minutes to 1 hour on a single GPU, depending on the transcript length.

Table 5 shows the average size of the search space (number of variables to optimize), along with Word Error Rate (WER) and EM score of the experiments. The EM score of GM without BoW is only 0.018, which indicates that the reconstruction fails for most utterances. However, GM with BoW can successfully reconstruct more than 50% utterances at the first run, and close to 100% utterances after 5 runs. Note that given BoW, in many cases, it is not a difficult task to rearrange the words to form a meaningful sentence. Our experiments, however, do not rely on language model guidance, which demonstrates that reconstruction succeeds with even uncommon word combinations, posing a security threat on utterances containing secret sequences.

We also perform reconstruction at different training stages. From 402 sampled utterances, we sample a smaller subset of 56 utterances. We train the ASR model and keep track of checkpoints at 0, 1k, 2k, 4k, and 10k-th step. Figure 2 shows the EM scores, and WER of the reconstructed transcripts. While GM on models at later training stage is less efficient, we can still reconstruct the transcript exactly for around 40% of the utterances.

# 5  Potential Negative Societal Impacts

As discussed in Section 2.1, our method can be potentially used by an adversary that compromises some client-to-server communication channel, or by an honest-but-curious host with access to a model update. The applications range from verifying if a sample of a certain class has been used to compute the update, to reconstructing a full sequence of labels used for training. Being aware of the existence of the techniques, a more secure system could be designed for the privacy of training participants.

# 6  Conclusion

We propose RLG, a method to reveal labels from a weight update in distributed training. We demonstrate that RLG can be applied to a wide variety of deep learning architectures, and can be used in conjunction with other methods to improve their performance. We also discuss the defensive effects of gradient compression techniques on the success of our method.

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
