# OpenReview forum: "Revealing and Protecting Labels in Distributed Training"
_NeurIPS.cc/2021/Conference — NeurIPS 2021 Poster_

### Official Review · Reviewer_u2De · 2021-06-25

**Rating:** 4
**Confidence:** 4

**Summary:**

The work shows that a set of labels used in a small batch of the training procedure is identifiable given the gradient of the last layer of the NN model with CE loss and soft-max activation. Also, they propose to modify gradients for defense.
There are some theoretical and empirical gaps between this work and real-world settings.

**Limitations And Societal Impact:**

addressed

**Main Review:**

Strengths:
- the work generalizes the recovery scenarios to multiple samples and steps, compared with a single sample and step solution by an earlier pioneering work
- the logic is clear

Weakness:
- It seems that the label not used in updating step also has the probability to be identified as "used". it is critical to provide rigorous proof for their proposed method.
- As the authors admit, their method does not work when the batch size is greater than the number of classes, which might be a major concern for application.
- The notation is unorganized and there are many typos that cause unnecessary troubles for understanding, e.g., the main result Remark 4.

**Time Spent Reviewing:**

2 hours

---

> ### Author Response · Authors · 2021-08-10
> **Response**
>
> Thank you for your time reviewing our work. We hope that following clarifications will help address your concern about the gap between this work and the real-world setting.
>
> 1. In this paper, we provide a necessary but not sufficient condition to identify labels included in the batch. We have seen in experiments that the necessary condition provides a reasonably good criterion in practice, giving high F1 scores in different scenarios. While our method can always return positive labels, it could also return false positives, which can be filtered out with additional information (e.g., knowledge of current model’s state for Gradients Matching), that we have demonstrated in Section 4.
> We also believe that a theoretical analysis would require deeper understanding to derive. For a certain point cloud (of columns in Q), it is possible that every point can be separated from the rest by a hyperplane passing through the origin. The distribution of points depends on the input to the last layer, which is hard to model, considering the wide variety of model architectures, training stages and scenarios that our method applies to. While it is interesting to theoretically understand its behavior, we demonstrate that our method works well in different practical scenarios. Note that there exist several recent works that empirically demonstrate the success of their proposed attacks  [Zhu et al’19, Geiping et al’20]
>
> 2. Prior works usually deal with batch sizes of less than 100 [Zhu et al’19, Yin et al’21]. In practice, when the method is applied to a gradient from an edge device, the batch size is usually small due to computational constraint and limited amount of data being stored for a limited period of time [Sim et. al‘19 (arXiv:1909.06678), Section 3.1].. For a practical scenario, our work shows that gradients sent from a user’s device to a training server can be used to reconstruct private image labels and text content even though raw data never left the device. Although the gradients may be aggregated across a large number of clients before being used to update the model, a small-batch gradient computed from a single user’s data [McMahan’16 (arXiv:1602.05629)] is accessible by the orchestrating server. We want to aim for protection to an extent that it is even hard for the orchestrating server to get access to user’s data.
>
> 3. We thank you for pointing this out. We will review Section 2.3, especially our remarks in the final version of the paper.

---

### Official Review · Reviewer_scvf · 2021-07-16

**Rating:** 6
**Confidence:** 1

**Summary:**

The paper proposes a new approach for revealing the training labels from the gradients of the last layer. The approach relies on the fact that if a label is used for computing the weight update, its corresponding column in the transposed right singular matrix can be linearly separated from other columns. The paper also experimented with how current gradient compression techniques can affect the attack accuracies, and explored how to reveal the sequence of labels, by combining the proposed approach with gradient matching.

**Limitations And Societal Impact:**

The authors discussed the limitations and potential negative societal impact of their work adequately.

**Main Review:**

I do not work in this area and I am not familiar with related work. I am not able to judge the originality and significance of the work. The following questions/suggestions are mostly on writing and clarity.

* Related work is discussed both in Section 1 and Section 2.4. Maybe it is better to put them in one place.
* In Section 2, the problem is defined in a way that the adversary only has access to some model updates. However, the approach in Section 4 needs more information than that (e.g., model weights). This makes the paper less coherent.
* The approach in Section 2 only needs the update of the weight matrix in the last layer. From a research standpoint, I understand that it is interesting to explore how we can use less information to extract what we want. However, I am curious how common it is in practice that the adversaries only have access to the update of the weight matrix in the last layer? In other words, if in practical scenarios, the adversaries have more information than that (e.g., the updates of all model weights), how your approach can be improved? Maybe it is useful to briefly discuss this in the paper.
* Line 112: the learning rate assumption does not work for common optimizers which have different adaptive learning rates for different values in the weight matrix (e.g., Adam).
* Eq 3: since softmax also depends on other values, it might be better to change the notation to explicitly point out which zs are used to in softmax normalization.
* Remark 4: q (the columns of Q) is not defined in the main text (only illustrated in Figure 1).
* Remark 4: from what I understand, q^c being linearly separatable from other columns does not necessarily ensure the existence of label c. In other words, the proposed algorithm could generate false positives. Is this correct? If so, this should be discussed in the paper.
* Besides the above, what other factors contribute to the drop of recall and precision? You mentioned in the experimental sections that the estimation of S could be inaccurate. Please discuss if there are other sources of errors, and show how much each of them contributes to the total errors.
* Does gradient matching (GM) applicable to the experiments in Section 2.5.1? And if so, why don't you compare with it? I see that you compared with GM in Section 2.5.2 and Section 4 only.
* Line 230: It mentioned that the prediction of S is inaccurate and therefore you use the ground-truth S for this experiment. This is inconsistent with the problem setup where you do not assume to know S. Please also show the results with the estimated S.
* Table 1: it is hard to compare the two algorithms as the corresponding numbers are not listed together.
* The current structure of the paper is attack (Sec. 2) -> defense (Sec. 3) -> attack (sec. 4). This is not a natural flow. Maybe you can consider reordering the sections.

**Time Spent Reviewing:**

4 hours

---

> ### Author Response · Authors · 2021-08-10
> **Response**
>
> Thank you for your helpful questions and comments towards improving our work. We number your questions and comments and address them one by one as below
>
> 2. Section 4 is an application of our method when used in conjunction with an existing technique and additional assumptions. The assumption that the adversary only has access to some model updates only applies to our method in Section 2.
> 3. While knowledge of other layers’ weights is not helpful to our method, it could be helpful to other techniques (e.g., Gradients Matching). We agree that it is uncommon for an adversary to know only about the last layer but not other layers. Rather than describing a practical attacking scenario, we provide a minimal condition for our method to work. We will add a discussion clarifying this point in the paper.
> 4. This is correct, and Adam, if applicable to this training paradigm, can provide a protection against our method. However, SGD remains the most common technique for distributed learning, especially federated learning (FedAvg, McMahan et al’17), whereas Adam and other optimizers need to be adapted to distributed settings [Reddi et al’20]. We will clarify this in the main text of the paper.
> 6. $q^i$ is meant to be i-th column in $Q$. We will clarify this in the text; thank you for pointing it out.
> 7. This is correct, we provide a necessary but not sufficient condition for the label to be included in the batch. It does not guarantee an accurate prediction. However, our results show that the rate of false positives is fairly small.
> 8. Analytically, our method has the recall of 1. However, in practice, error comes from precision errors of SVD, linear programming, and perceptron algorithms, since they do not always give a precise solution, especially when the input values are small (e.g., gradients at a late training stage).
> 9. Yes, GM can be used in 2.5.1. However, this requires reconstructing the whole batch of image samples for the labels to also be recovered. In fact Yin et al’21 use the baseline to reconstruct labels first, then fix the labels and reconstruct image samples. We believe they do that because the results are better than using GM alone.
> 10. We agree with you that S should be estimated in these experiments to be consistent with the problem setup and other experiments. However, while we can predict S correctly in most cases involving a single step, we find that S is hard to predict in the multi-step setting, due to the precision error. We hypothesize that in the multi-step single-sample case, the gradient at each step is too similar that they are linearly dependent, making estimation for the rank of $\nabla W$ erroneous. We use SVD to estimate the rank of a matrix, which requires specifying a threshold epsilon (10^-5 in most experiments) to cut off non-zero eigenvalues. While the error of batch size (or sequence length in this case) is almost zero in single-step cases, it is, for example, 8.18, in the case of 8-step. Since we do not want to go into the complication of tuning this epsilon value, we report the results with ground-truth value as to just demonstrate the technique working with more complicated scenarios. More engineering would be involved to get the optimal results. For the final version of the paper, we will include length estimation errors.
>
> 1, 5, 11, 12. We agree with your comments and will consider incorporating them into the final version of the paper.

---

> > ### Comment · Reviewer_scvf · 2021-08-19
> > **Thank you**
> >
> > I thank the authors for the response, which answers all my questions.

---

### Official Review · Reviewer_6fay · 2021-07-16

**Rating:** 6
**Confidence:** 4

**Summary:**

The primary objective of paper is to propose a method which can reveal labels only from shared gradient information in distributed learning methodologies.

**Limitations And Societal Impact:**

The paper shows a mathematical approach using SVD to predict labels using gradients from final layers in neural network, while this is a novel approach a few improvements can be recommended:

1. Security flaws in federated learning (and split learning) are well known and active research is being done on mitigation. However authors only look at the vanilla gradient sharing approach and then talk about mitigations. As [1] proposes there are methods including gradient noise to mitigate such flaws. This limits the impact of paper. and paper would benefit from analysis of leakage when security measures in papers like [1] are being applied.
2. Most security work is focused around reconstruction of data and not labels - there is a reason for that. Raw data contains important information about privacy of individuals. While labels are crucial too - its easy to obfuscate the labels and project them in a different embedding - A second crucial factor about impact of a label recovery focused method.
3. The paper talks about revealing labels in distributed learning. However authors only address it for federated learning. There are other distributed learning methodologies [1, 2] which should be both referenced in related work and studied from point of view of security.
4. While multiple architectures and datasets are used - the paper doesnt report variance numbers / error metrics around the prediction - making it hard to assess the robustness of method from experiments alone.
5. The paper is difficult to understand - exposition could be simplified.



[1] Vepakomma, Praneeth, et al. "NoPeek: Information leakage reduction to share activations in distributed deep learning." 2020 International Conference on Data Mining Workshops (ICDMW). IEEE, 2020.

[2] Gupta, Otkrist, and Ramesh Raskar. "Distributed learning of deep neural network over multiple agents." Journal of Network and Computer Applications 116 (2018): 1-8.



**Main Review:**

The paper shows a clever mathematical derivation of how gradients are connected to labels - as a product of two row matrices h,g. the matrices themselves are derivative of loss function and hidden layer activations - and linear programming and SVD based methods can be used to identify the labels under certain conditions. the paper demonstrates results on ASR and Imagenet. the paper suffers from certain limitations - overlooking other methods in distributed learning and just focusing on federated learning, only looking at label reconstruction, only looking at vanilla version of federated learning, not reporting variance on metrics reported - and fundamental limitation of algorithm itself S < min{d, C}

**Time Spent Reviewing:**

2

---

> ### Author Response · Authors · 2021-08-10
> **Response**
>
> Thank you for your time reviewing our work. We appreciate your recommendations for improvement of this work and want to address your concerns about these limitations.
> 1. While leakage from gradients has already been investigated, many mitigations come at a cost to utility (e.g. DL with differential privacy [Abadi et al’16]) [Zhu et al’19, Dang et al’21], while some prior-used defenses (e.g., Dropout) are not applicable against this attack. While [1] proposed an interesting technique on defending distributed learning models, we believe that it does not work in our settings, since the leakage comes from gradients. Although we apply the method on vanilla gradient sharing as the default setting, we do demonstrate its effect on gradient compression and sparsification from the perspective of defense methods.
> 2. We agree that in some scenarios, label reconstruction is less interesting than the raw data. Since a forward and backward pass needs to be performed on the client side with data and labels from the user, a label obfuscation could be invertible. The client can choose to freeze the last feed-forward (or projection) layer, thus not providing its gradient, but training this feed-forward layer is the current standard in a majority of deep learning models.
> 3. While we are aware of different distributed learning techniques, many production-scale NNs are trained via FL (e.g. mobile keyboard prediction, Hard et al’18). We focus on FL since it is an increasingly popular framework with gold standard privacy notions like differential privacy (e.g., McMahan et al’17 (arXiv:1710.06963), Thakkar et al’20 (arXiv:2006.07490)).  Although the experiments mainly focus on FL, we will add a discussion that the technique is not restricted to FL. We also make sure to include a discussion of other common distributed paradigms as you mentioned in the related works section in the final version of the paper.
> 4. While reporting variance is useful, we note that our method is very effective on a random and reasonably large subset of samples. For example, in Table 1, the results are averaged across 100 batches. Since the reported F1 is close to 1, the variance must be very small. We will add variance estimates for all feasible experiments, including those in Section 2.
> 5. Thanks for pointing this out. We will carefully revise the draft and simplify our expositions wherever possible.

---

> > ### Comment · Reviewer_6fay · 2021-08-20
> > **Satisfactory**
> >
> > Thanks for responding to all my questions. Additionally the responses to 1, 2 need to be reflected somewhere in paper - perhaps on discussions, conclusions.

---

> > > ### Author Response · Authors · 2021-08-23
> > > **Thank you for the helpful comments**
> > >
> > > We will add text in the relevant sections in the paper to appropriately reflect the responses to 1,2. Thank you!

---

### Official Review · Reviewer_Bkhg · 2021-07-23

**Rating:** 7
**Confidence:** 2

**Summary:**

This paper presents a method, RLG (Revealing Labels from Gradients), from only the gradient of the last layer and the id to label mapping. The method is tested on two domains: (1) image classification; and (2) automatic speech recognition. The results indicate that under a reasonably small batch size, the proposed method is robust to a wider range of data sets while the baseline performs well only on a smaller subset. The paper next shows that gradient quantization and gradient sparsification could be possible defense strategies against the proposed method.

**Limitations And Societal Impact:**

Yes.

**Main Review:**

The paper presents a new method to uncover potential vulnerabilities in distributed learning systems.
The method, its performance and possible defense strategies are described well.

Questions:

1.  How much accuracy can be lost when the defense strategies are used?

The paper is written well.

One minor typo: a honest-but-curious  -> an honest-but-curious


**Time Spent Reviewing:**

4

---

> ### Author Response · Authors · 2021-08-10
> **Response**
>
> Thank you for your time reviewing our work. As you pointed out that our method works with a reasonably small batch size due to the technical limitation of S (batch size) < min{d, C}, note that d and C are usually large (in order of hundreds to thousands) in many production-scale model architectures, and S is usually small when distributed learning is performed from edge devices.
>
> To answer your question, while we focus on the mitigation benefits of the gradient compression and sparsification techniques, these have been shown in prior works [Seide et al’14, Aji et al’17] to provide a communication-efficient distributed learning paradigm without affecting the accuracy.

---

### Decision · Program_Chairs · 2021-09-27

**Decision:**

Accept (Poster)

**Comment:**

This paper gives an interesting way to reveal label information from the transmitted gradients in distributed training. Empirically, the paper also shows that techniques for communication reduction mitigate the privacy leakage of label information. Overall, this paper gives new insights into what information can be leaked from the model updates in federated learning. The authors should incorporate the reviewers' suggestions in the next revision.